# Unidirectional ray polaritons in twisted asymmetric stacks

J. Álvarez-Cuervo [1,2,14], M. Obst [3,4,14], S. Dixit [5,14], G. Carini[6], A. I. F. Tresguerres-Mata [1,2], C. Lanza [1,2], E. Terán-García [1,2], G. Álvarez-Pérez [1,2,6,7], L. F. Álvarez-Tomillo [1,2], K. Diaz-Granados[5], R. Kowalski[5], A. S. Senerath[5], N. S. Mueller[6], L. Herrer [8], J. M. De Teresa [8], S. Wasserroth[6], J. M. Klopf [9], T. Beechem [10], M. Wolf [6], L. M. Eng [3,4], T. G. Folland [11], A. Tarazaga Martín-Luengo [1,2], J. Martín-Sánchez [1,2], S. C. Kehr [3,4] ✉, A. Y. Nikitin [12,13], J. D. Caldwell [5] ✉, P. Alonso-González [1,2] ✉ & A. Paarmann [6] ✉

The vast repository of van der Waals (vdW) materials supporting polaritons offers numerous possibilities to tailor electromagnetic waves at the nanoscale. The development of twistoptics—the modulation of the optical properties by twisting stacks of vdW materials—enables directional propagation of phonon polaritons (PhPs) along a single spatial direction, known as canalization. Here we demonstrate a complementary type of directional propagation of polaritons by reporting the visualization of unidirectional ray polaritons (URPs). They arise naturally in twisted hyperbolic stacks with very different thicknesses of their constituents, demonstrated for homostructures of $\alpha$-MoO$_3$ and heterostructures of $\alpha$-MoO$_3$ and $\beta$-Ga$_2$O$_3$. Importantly, their ray-like propagation, characterized by large momenta and constant phase, is tunable by both the twist angle and the illumination frequency. Apart from their fundamental importance, our findings introduce twisted asymmetric stacks as efficient platforms for nanoscale directional polariton propagation, opening the door for applications in nanoimaging, (bio)-sensing, or polaritonic thermal management.

In recent years, the exploration of PhPs in polar materials has emerged as a promising avenue in nanophotonics, offering remarkable control over electromagnetic waves at the nanoscale[1–4]. The applications of PhPs are diverse, encompassing molecular sensors[5,6], hyperlensing[4,7–9], enhanced thermal emission[10], and detectors[11], among others. To engineer nanophotonic devices effectively, it becomes imperative to manipulate the characteristic features of PhPs, such as their wavelength, propagation direction, propagation length or lifetimes.

[1]Department of Physics, University of Oviedo, Oviedo, Spain. [2]Center of Research on Nanomaterials and Nanotechnology (CINN), CSIC-Universidad de Oviedo, El Entrego, Spain. [3]Institute of Applied Physics, TUD Dresden University of Technology, Dresden, Germany. [4]Würzburg-Dresden Cluster of Excellence —EXC 2147 (ct.qmat), Dresden, Germany. [5]Vanderbilt University, Nashville, TN, USA. [6]Fritz Haber Institute of the Max Planck Society, Berlin, Germany. [7]Center for Biomolecular Nanotechnologies, Istituto Italiano di Tecnologia, Via Barsanti 14, Arnesano, Italy. [8]Instituto de Nanociencia y Materiales de Aragón (INMA), CSIC-Universidad de Zaragoza, Zaragoza, Spain. [9]Institute of Radiation Physics, Helmholtz-Zentrum Dresden-Rossendorf, Dresden, Germany. [10]Purdue University and Birck Nanotechnology Center, West Lafayette, IN, USA. [11]University of Iowa, Iowa City, IA, USA. [12]Donostia International Physics Center (DIPC), Donostia-San Sebastián, Spain. [13]IKERBASQUE, Basque Foundation for Science, Bilbao, Spain. [14]These authors contributed equally: J. Álvarez-Cuervo, M. Obst, S. Dixit. ✉e-mail: susanne.kehr@tu-dresden.de; alexey@dipc.org; josh.caldwell@vanderbilt.edu; pabloalonso@uniovi.es; alexander.paarmann@fhi-berlin.mpg.de

Notably, it has been demonstrated that the fundamental properties of PhPs can be engineered via the choice of the symmetry of the host crystal[12]. For instance, while PhPs in uniaxial crystals such as hexagonal boron nitride (hBN) exhibit isotropic in-plane and hyperbolic out-of-plane propagation[7,8,13,14], lower symmetry crystals like $\alpha$-MoO$_3$ or $\alpha$-V$_2$O$_5$ can support both in- and out-of-plane hyperbolic propagation[15–19], due to their biaxial nature. Typically, the propagation of polaritons for an in-plane hyperbolic medium is dominated by the low-momentum components at the base of the hyperbolic Iso-frequency Curve (IFC) – a planar cut through the polariton dispersion at a constant frequency[15–19]. However, if the propagation is dominated by the asymptotes of the hyperbola, ray-like polaritons emerge along a single spatial direction with a constant phase, which results from the momentum **k** being almost entirely perpendicular to the direction of propagation defined by the Poynting vector **S**. Besides, these polaritons are highly confined, strongly bound to the interface of the host material, and contain a high density of electromagnetic states[20]. As a result, ray polaritons can, for example, preserve high resolution, transmitting large-k information over large distances without diffraction[21], and thus providing a distinctive opportunity for optoelectronic applications that rely on nanoscale waveguiding and steering.

Recent studies have successfully visualized in-plane ray-like polariton modes. For instance, two PhPs rays with a cross-like shape have been observed at the surface of an $\alpha$-MoO$_3$ layer placed over a SiC substrate[22]. A similar effect, but with a pronounced asymmetry between the intensity of the two rays has been demonstrated in crystal off-cuts of calcite[21,23] and in monoclinic crystals like $\beta$-Ga$_2$O$_3$ and CdWO$_4$[24–27]. The asymmetry between these two rays can be so large that, effectively, only one of the rays is observed. However, in these studies[21,23–27] the unidirectional ray propagation can only be achieved using specifically designed polariton launchers or concrete incident polarization conditions that allow for control over which components of the IFCs are excited. Thus, in these systems, URPs do not emerge naturally from the material properties.

The possibility of combining layers of van der Waals (vdW) materials in a single twisted stack expands the potential for engineering and customizing the PhPs dispersion, enabling precise control over their propagation characteristics (known as twistoptics[12,28–34]). As a hall-mark feature, canalized PhPs—waves propagating along a single spatial direction—have been observed in twisted homostructures of thin $\alpha$-MoO$_3$ layers[30–35]. By including a third layer of similar thickness, it is even possible to guide these canalized polaritons along any desirable in-plane direction[34]. This tunability is crucial to efficiently apply these unidirectional PhPs to a wide range of nanophotonic applications. It is important to note that although both unidirectional ray-like and canalized polaritons propagate along a single spatial direction, defined by a unique direction of the Poynting vector[30–34], they have clearly distinct propagation behavior: canalized polaritons contain momentum components from many (ideally all) directions of the in-plane momentum space, resulting in periodic amplitude oscillations along the direction of propagation. In contrast, URPs exhibit momentum components only from a single direction perpendicular to the direction of propagation, resulting in a non-oscillatory propagation exhibiting a constant phase.

Here, we report the experimental observation of URPs being naturally supported in twisted asymmetric structures composed of a thin and a thick hyperbolic layer. These URPs exhibit a high degree of tunability by changing either the illumination frequency or the twist angle, i.e., without the need for specific antenna designs or illumination conditions. In particular, we observe URPs in two complementary systems: a twisted homostructure comprising identical hyperbolic materials for each layer and a twisted heterostructure featuring different hyperbolic materials with strongly asymmetric layer thicknesses in both systems. Consequently, only one thin vdW layer is required to fabricate these stacks, which constitutes a significant simplification

with respect to previous reports of unidirectional PhP propagation where double or triple thin layers were used. The emergence of URPs in both homo- and heterostructures demonstrates the versatile nature of the phenomenon, thereby advancing our understanding and broadening the applicability of twist-optical platforms.

## Results

### Emergence of unidirectional ray polaritons

Schematics of the two systems are shown in Fig. 1a, b: a thin $\alpha$-MoO$_3$ layer is placed on a thick $\alpha$-MoO$_3$ layer to form the homostructure (Fig. 1a), or on a thick layer of $\beta$-Ga$_2$O$_3$ to form the heterostructure (Fig. 1b). Parameters $\theta_1$ and $\theta_2$ represent the corresponding twist angles between the thin and thick layers for both structures. Throughout this work, we will consider that the coordinate axes are aligned to the main crystal axes of the top layer; i.e., the x (y,z) axis is fixed to the [100] ([001], [010]) direction of the top $\alpha$-MoO$_3$ layer. Particularly, changing the twist angles must be understood as a rotation of the bottom layer with respect to the [100] direction of the $\alpha$-MoO$_3$ top layer. Examples of the polaritonic response expected in these systems are shown in Fig. 1c–h, where real-space numerical calculations (see Methods) of the polaritonic field Re(E$_z$) produced by a point dipole at the top surface of the thin $\alpha$-MoO$_3$ layer are displayed, along with its 2D Fast Fourier Transform (2D-FFT) shown in the insets.

For both systems, there exists a specific twist angle $\theta$ for which PhPs propagate with a cross-like shape (Fig. 1c, f). The observed dual-directional ray-like propagation is clearly distinct from the hyperbolic propagation, where several fringes with a hyperbolic shape propagate far from the source. Yet, mirror symmetry with regard to both [100] and [001] $\alpha$-MoO$_3$ crystal axes, typical for polaritonic response in single layers of orthorhombic biaxial materials, is conserved for these cases. We, therefore, term this condition the symmetrical case. Remarkably, these mirror symmetries are broken when considering different twist angles (Fig. 1d, e, g, h). In particular, the polariton ray along one direction gradually increases its electromagnetic field intensity and propagates further when varying the twist angle between the layers. For the rays along the other direction, the opposite effect occurs: their intensity is reduced, and the rays become shorter until they almost disappear. Through a gradual adjustment of the twist angle, we can maximize both effects. At an optimal twist angle, only one branch gives rise to polariton propagation, while the other branch is suppressed almost completely (Fig. 1e, h). Therefore, unidirectional propagation of PhPs emerges naturally in these systems, constituting a perfect platform for the development of applications where a strong directional control of light at the nanoscale is required. Remarkably, the PhPs excited in these extremely anisotropic cases exhibit a ray-like behavior; in other words, the direction of propagation is perpendicular to the wavefronts, which present a constant phase.

This polariton propagation in real-space corresponds to a linear, unidirectional behavior also in the momentum space representation (insets of Fig. 1e, h), where the majority of polariton momentum components point in a single direction. This effect is in stark contrast to previous observations of unidirectional propagation of canalized PhPs in twisted bilayers and trilayers of $\alpha$-MoO$_3$[30–34]. There, employing twisted thin layers of similar thickness results in linear IFCs that are offset from the origin, and thus contain omnidirectional momentum components. From a fundamental point of view, the unidirectional linear IFCs introduce a variety of unexplored phenomena in which reflection[36–38] and refraction[37–41] are prominent examples. Notably, the direction of propagation of both rays in the homostructure rotates gradually with the twist angle. This effect becomes clear when extracting the $\varphi_1$ value, which represents the direction of the twist-enhanced ray with respect to the [100] top layer direction. We extract $\varphi_1 = 36°$, $\varphi_1' = 50°$, and $\varphi_1'' = 63°$ from Fig. 1c, d, e, respectively. Thus, changing the twist angle between the layers not only enhances the

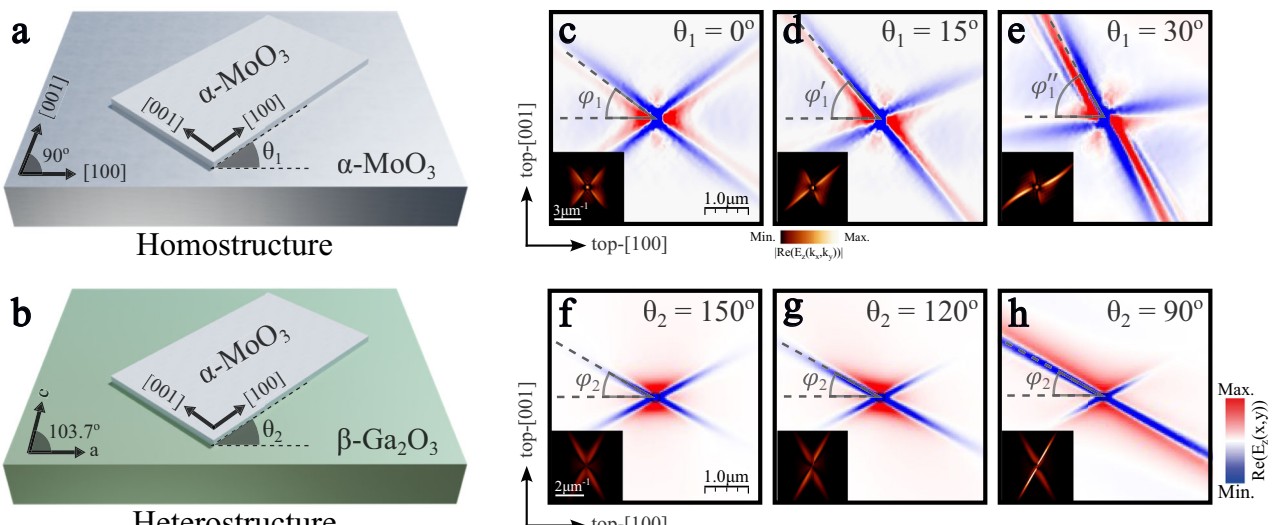

**Fig. 1 | Unidirectional ray polaritons in twisted asymmetric stacks.** Schematic of the two systems under study: **a** a 80 nm-thin $\alpha$-MoO$_3$ layer placed over a 3 $\mu$m-thick $\alpha$-MoO$_3$ layer (homostructure) and **b** a 100 nm-thin $\alpha$-MoO$_3$ layer on top of a 5 $\mu$m-thick $\beta$-Ga$_2$O$_3$ layer (heterostructure). Parameters $\theta_1$ and $\theta_2$ represent the twist angle of the thick bottom layer with regard to the thin top layer for both systems, respectively. Real-space electric fields Re(E$_z$) launched by a point dipole source at the thin $\alpha$-MoO$_3$ surface for the homostructure (**c**–**e**) and the heterostructure (**f**–**h**) at illumination frequencies $\omega = 920$ cm$^{-1}$ and 734 **cm$^{-1}$**, respectively. The twist angles are $\theta_1 = 0^{\circ}$ (**c**), 15$^{\circ}$ (**d**), 30$^{\circ}$ (**e**) and $\theta_2 = 150^{\circ}$ (**f**), 120$^{\circ}$ (**g**), 90$^{\circ}$ (**h**). The in-plane direction of ray-like propagation is marked with dashed gray lines at an angle defined by $\varphi_1$, $\varphi_1'$, and $\varphi_1''$ (**c**–**e**) and $\varphi_2$ (**f**–**h**) with respect to the [100] crystal direction of the top $\alpha$-MoO$_3$ layer. The Fourier Transforms (2D-FFTs) of the simulated real-space images are shown in the insets of **c**–**h**. The same scale has been used for every simulated real-space map.

amplitude of that ray but also rotates the whole cross by a similar angle.

The behavior of the heterostructure is very similar in general, however, also exhibits some differences. Firstly, the symmetric case (Fig. 1f) emerges at a non-trivial twist angle due to the frequency-dependent optical axis direction of $\beta$-Ga$_2$O$_3$[24]. Upon twisting (Fig. 1g, h), the intensity ratio between both ray directions change similarly to the homostructure behavior, while for the heterostructure additionally also the propagation lengths is modulated (see Supplementary Fig. 15). However, and in contrast to the homostructure, the direction of the enhanced ray, defined by $\varphi_2$, remains constant independent of the twist angle of the thick $\beta$-Ga$_2$O$_3$ layer. Consequently, although both asymmetric structures support unidirectional polariton rays, there are also distinct differences in their behavior, suggesting that the physical reason for unidirectional ray propagation may be different. Hence, the two structures constitute an ideal platform for the study of the complex generation of PhPs in twisted bilayer systems with vastly different layer thicknesses.

**Unidirectional ray polaritons in twisted homostructures**

To demonstrate experimentally the existence of URPs in twisted homostructures, we first fabricated a stack formed by a thin $\alpha$-MoO$_3$ layer ($d_{top} = 80$ nm) over a thick $\alpha$-MoO$_3$ layer ($d_{bot} = 3$ $\mu$m) with twist angles $\theta = 30^{\circ}$ and 60$^{\circ}$ (see Methods). The propagation of PhPs (launched with the help of a 200-nm diameter hole fabricated by focused ion beam milling) was visualized by scattering-type scanning near-field optical microscopy (s-SNOM, see Methods). The image obtained for the case of $\theta = 30^{\circ}$ and $\omega = 880$ cm$^{-1}$ (Fig. 2a) shows a bright polaritonic ray that propagates away from the hole with decaying amplitude along the in-plane direction defined by $\varphi = 30^{\circ}$. Parallel to this fringe, a second dark fringe is formed, revealing that the polaritonic wavefront is completely perpendicular to the 30$^{\circ}$ direction of propagation. This observation constitutes, as previously described, the fingerprint of unidirectional ray-like propagation. To further corroborate this result, we perform a 2D-FFT on the experimental image. The resulting IFC (inset in Fig. 2a) shows a linear shape, demonstrating

the existence of only one allowed direction for both the polaritonic wavevector and the energy flow (perpendicular to the wavevector), as expected for a ray.

To investigate the spectral response and potential tunability of URPs in the twisted homostructure, we measured other illumination frequencies within the [100] hyperbolic $\alpha$-MoO$_3$ Reststrahlen band[14–17], as shown in Fig. 2b, c for $\omega = 900$ cm$^{-1}$ and 920 cm$^{-1}$, respectively (see Supplementary Figs. 1 and 2 for other frequencies). The experimental images obtained show unidirectional ray-like propagation of PhPs similar to Fig. 2a; however, the direction of the ray varies with frequency. In fact, as illustrated in the s-SNOM images (dashed lines), the angle between the direction of propagation of the PhPs and the [100] top layer direction, $\varphi$, undergoes a change of up to 30$^{\circ}$ between $\omega = 880$ cm$^{-1}$ and $\omega = 920$ cm$^{-1}$. This effect can also be appreciated by observing the evolution with frequency of the slope of the linear IFCs in the FFT. This frequency behavior enables remarkable control over light propagation at the nanoscale. Despite numerous studies in twis-toptics successfully demonstrating control over the directionality of light and the polaritonic response[30–34], a common challenge across these works stems from the need for mechanical twist angle variation between layers. Here instead, we demonstrate that the orientation of the PhP propagation can be tuned by simply changing the excitation frequency. Consequently, twisted asymmetric homostructures serve as ideal platforms for a range of potential applications, including routers or directional biosensors.

To corroborate these experimental results, we performed full-wave numerical simulations (Fig. 2e–g), showing excellent agreement with the s-SNOM near-field real-space images, both in terms of the shape of the polariton wavefront and its propagation direction. The simulations, as well as the experiments, are characterized by the presence of a single polariton ray and the almost complete absence of the other one, rendering these polaritons fully unidirectional. The simulations only show a very faint polaritonic feature close to the [100] direction corresponding to a weak polaritonic mode along the second asymptote in the IFC, which is almost completely suppressed (see insets). Twisting the two layers reduces the intensity of one of the

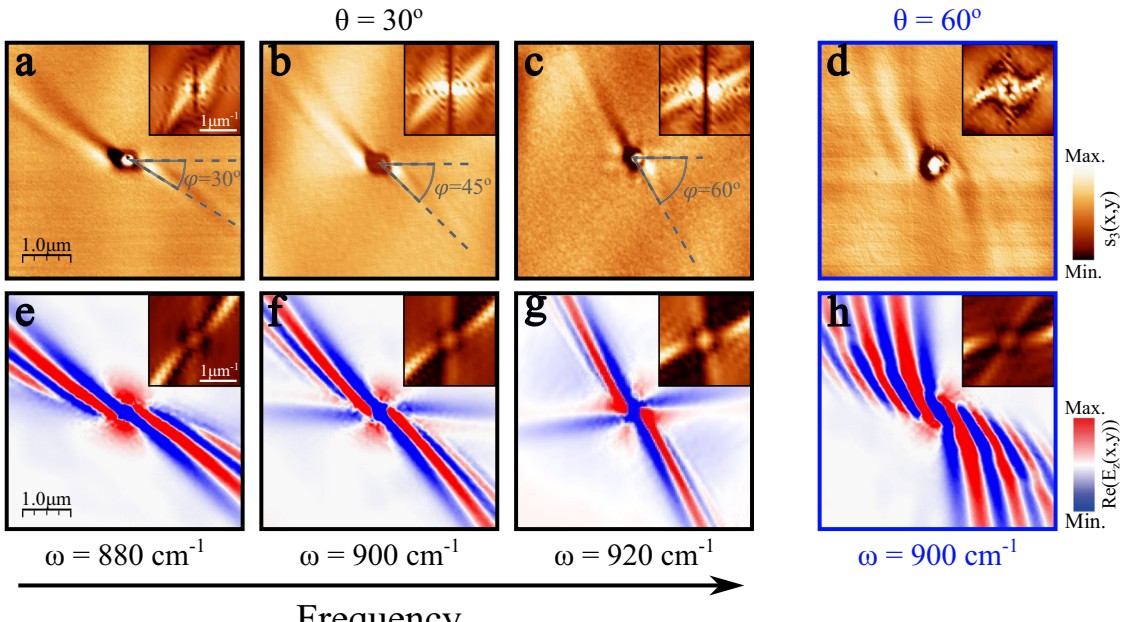

**Fig. 2 | Observation of unidirectional ray polaritons in twisted asymmetric homostructures.** Near-field amplitude image in a twisted structure made of a 80-nm thin $\alpha$-MoO$_3$ layer over a 3-μm thick $\alpha$-MoO$_3$ layer with twist angles $\theta = 30°$ (**a–c**) and 60° (**d**) at an illuminating frequency of $\omega = 880\ cm^{-1}$, 900 cm$^{-1}$, 920 cm$^{-1}$ and 900 cm$^{-1}$ for **a–d**, respectively. A 200-nm diameter hole allows efficient launching of the phonon polaritons, whose wavefronts and direction of propagation are visualized by s-SNOM (scattering-type scanning near-field optical microscopy). For **a–c** the in-plane direction of propagation is marked with gray lines at an angle defined by $\varphi$ with respect to the [100] crystal direction of the top $\alpha$-MoO$_3$ layer, see Fig. 1a. The experimental isofrequency curves (Fast Fourier Transform (2D-FFT) of the near-field image) are shown in the insets, verifying polariton unidirectional propagation in the direction defined by $\varphi$. **e–h** Simulated near-field amplitude images of the system in **a** (**e**), **b** (**f**), **c** (**g**) and **d** (**h**). The 2D-FFTs of the simulated images are shown in the top insets of **e–h**.

previously crossed rays towards its full suppression (Supplementary Fig. 6), enabling URPs. Remarkably, the angular distance between the enhanced and the suppressed ray increases with frequency. This is in excellent agreement with the frequency evolution of the IFCs for a single thick layer of $\alpha$-MoO$_3$ (Supplementary Fig. 3). Careful inspection of the FFTs of the experimental and simulated propagation patterns reveals that the IFCs are not perfectly linear but show a slight curvature. With increasing frequency, this curvature of the IFCs diminishes, resulting in a reduced number of fringes (Supplementary Fig. 3).

A more extreme PhP behavior can be observed for a twist angle of $\theta = 60°$ (Fig. 2d). An exotic "pinwheel" pattern arises naturally from the asymmetric homostructure. Several polariton fringes can be observed within a narrow angular sector. However, for this configuration, the phase fronts are curved, generating a completely different field profile compared to the URPs observed at the smaller twist angles. Both features can be clearly identified from the momentum space data (FFT in the inset in Fig. 2d). As before, one branch of the IFC is much stronger than the other, resulting in strongly unidirectional propagation, however, with a significant curvature of the IFCs consistent with the emergence of several curved fringes covering a narrow angular sector in real space. Again, the numerical simulation for $\theta = 60°$ (Fig. 2h) is in excellent agreement with the experimental result, matching both the directionality and periodicity of the fringes.

To conceptually understand why URPs emerge in the asymmetric homostructure, we performed numerical simulations (Fig. 3) of the transition from a symmetric $\alpha$-MoO$_3$ twisted bilayer (equal thicknesses of the layers) to the asymmetric homostructure. For consistency, we nominally maintain the thickness of the top $\alpha$-MoO$_3$ layer constant as $d_{top} = 80$ nm, and increase the thickness of the twisted bottom layer: $d_{bot} = 80$ nm, 500 nm, and 3 μm, keeping a twist angle of $\theta = 30°$ and an illumination frequency of $\omega = 900\ cm^{-1}$. A scheme of the three structures is shown in Fig. 3a–c. For the symmetric bilayer (Fig. 3a), a typical

hyperbolic PhP propagation[30–33], slightly tilted due to the coupling between the two layers, can be observed in the real-space images and the resulting IFCs (Fig. 3e, i). Increasing the bottom layer thickness to 500 nm (Fig. 3b) results in significant changes in the PhP propagation. Two distinct effects can be identified in the simulated near-field data (Fig. 3f); (i) an irregular hyperbolic pattern propagates close to the horizontal direction. Note that the intensity of these fringes decays more quickly than in the symmetric bilayer. (ii) The fringes become longer, slightly curved and more aligned along the diagonal. Both effects maximize for the thickest bottom layer (Fig. 3c). Here, the phase fronts flatten out completely and lose their hyperbolic shape. At this point, only the diagonal fringes persist, yet in a much flatter manner, generating the unidirectional ray-like propagation observed in Fig. 2b, f.

A fundamental explanation of the physical effect responsible for this behavior can be found in the momentum space representation (Fig. 3i–l). The white dashed line in Fig. 3i–k corresponds to the analytic IFC of the symmetric bilayer obtained with Equation (S1), previously derived in ref. 34. Remarkably, the IFCs for the thicker bottom layers (Fig. 3j, k) converge towards the IFC of the symmetric bilayer for high momentum values at a specific angular region defined by the white and orange lines in the panels. These lines correspond to the asymptotes of the IFC for the bottom and top layers, respectively, when considered alone. They are defined by the angle $\varphi$ at which the real part of the in-plane projected permittivity

$$\varepsilon_{MoO_3}^{\varphi} \equiv \varepsilon_{MoO_3,x}\cos^2\varphi + \varepsilon_{MoO_3,y}\sin^2\varphi \qquad (1)$$

vanishes, i.e., $\text{Re}\left[\varepsilon_{MoO_3}^{\varphi}(\varphi)\right] = 0$. Consequently, in the narrow cone between these two lines, the top layer has negative projected permittivity values whereas $\text{Re}\left[\varepsilon_{MoO_3}^{\varphi}(\varphi)\right] > 0$ for the bottom layer. Therefore,

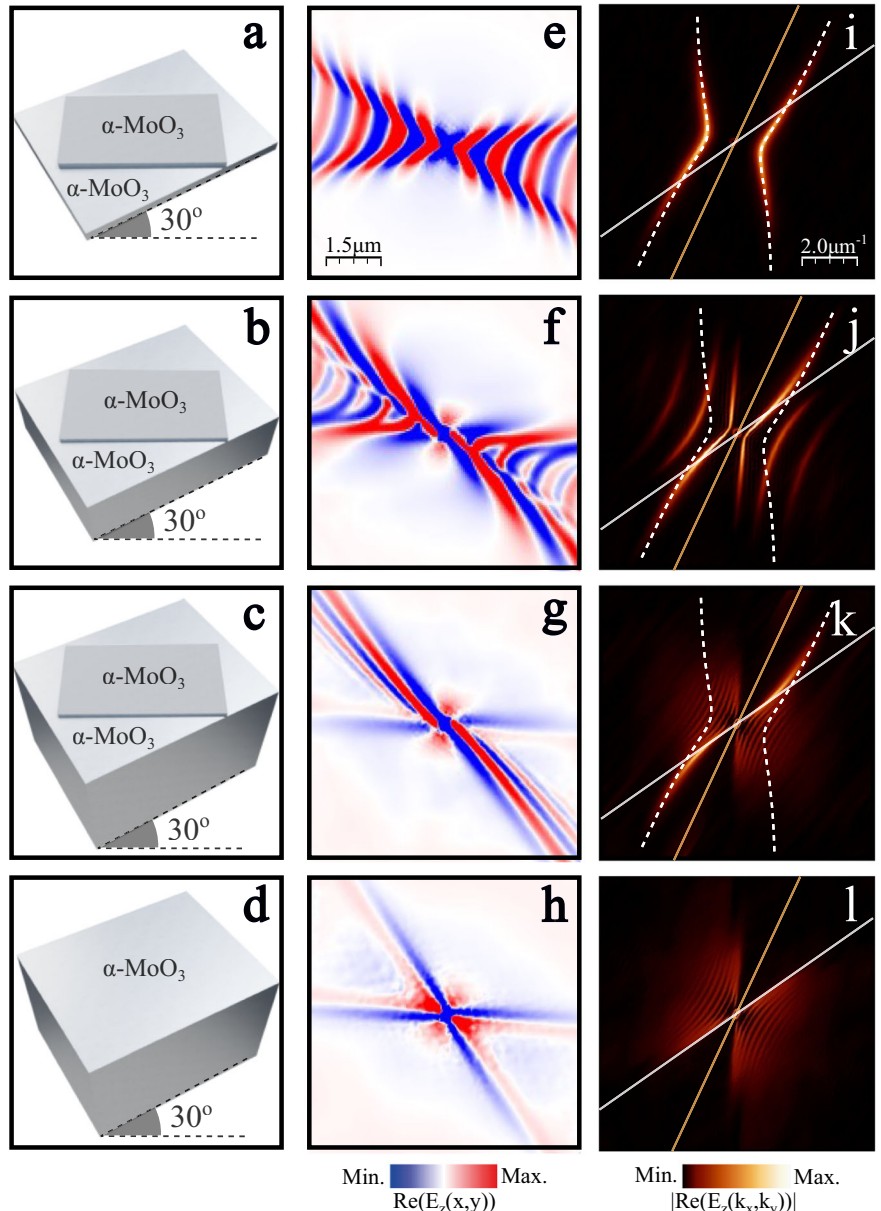

**Fig. 3 | Analysis of the thickness disparity in twisted homostructures.**
**a–d** Schemes of four systems made of $\alpha$-MoO$_3$ layers. The top thin layer in **a–c** has a thickness of $d_{top} = 80$ nm while the bottom has $d_{bot} = 80$ nm (**a**), $d_{bot} = 500$ nm (**b**), $d_{bot} = 3$ µm (**c**). Scheme **d** corresponds to a single thick layer of $\alpha$-MoO$_3$ ($d = 3$ µm). In all cases, the twist angle is $\theta = 30°$ and the illumination frequency is $\omega = 900$ cm$^{-1}$. **e–h** Numerical simulations showing the near-field Re(E$_z$) generated by a point dipole located above the four structures represented in **a–d**, respectively. **i–l** Isofrequency curves (IFCs) obtained by performing the Fourier Transforms (2D-FFTs) of the near-field images in **e–h**, respectively. White dashed curves in **i–k** correspond to the analytic IFC of the bilayer case shown in **a**. White and orange lines in **i–l** are the asymptotes of the IFCs of phonon polaritons in the bottom and top layers, respectively.

in this angular region the bilayer structure effectively supports a polariton mode confined to the top layer that is modified by a positive permittivity substrate. This holds in particular for large momentum components, where the mode decays rapidly into the bottom layer such that its finite thickness does not contribute (Supplementary Fig. 4). Hence, increasing the substrate thickness has little influence on the IFC at large momentum values. As a result, the IFC of the asymmetric homostructure tends towards the IFC of the symmetric bilayer for large momenta.

On the other hand, the part of the IFC in the symmetric bilayer that is outside the narrow cone between the white and orange lines belongs to the angular sector where the projected permittivity of the bottom layer is negative. This implies that in this momentum range each layer supports polariton modes. Increasing the bottom layer

thickness decreases the confinement of the respective modes, which translates into IFCs with smaller momentum values. This feature can be clearly observed at the IFC of the 500 nm-thick bottom layer (Fig. 3j). When the bottom layer is sufficiently thick (Fig. 3k), the polaritons in the full structure are dominated by the bottom layer modes with low confinement. For this momentum region, the wavelength of the bottom layer polariton approaches the free-space wavelength values and, thus, no interference with the thin top layer is expected. This becomes clear when finally removing the top layer and only considering the thick bottom layer (Fig. 3d). The short length-scale propagation pattern of this single thick slab (Fig. 3h) also shows a ray-like, yet symmetric, propagation pattern[20]. The corresponding IFC of this thick layer (Fig. 3l) resembles the IFC of the stacked homostructure in this region of momentum space. With all these elements, we can

$$\omega = 734 \text{ cm}^{-1}$$

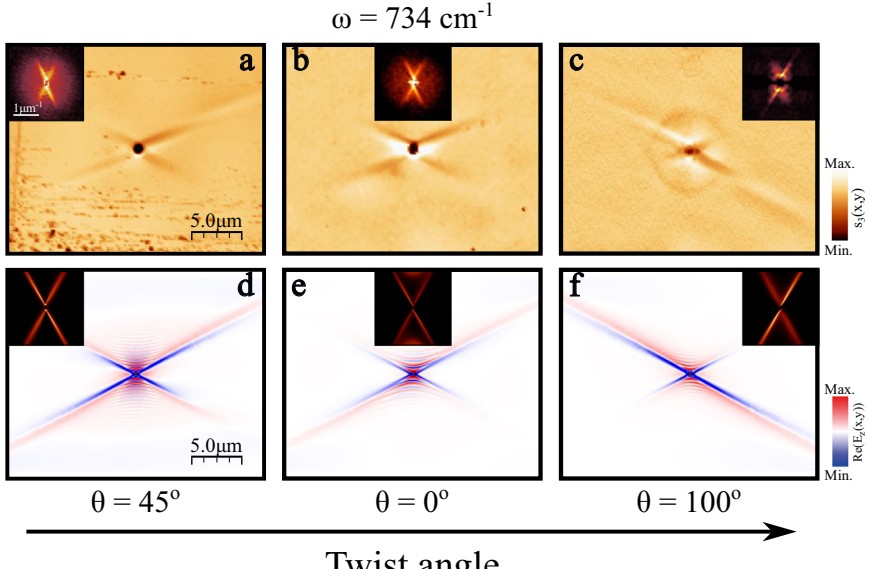

**Fig. 4 | Observation of unidirectional ray polaritons in twisted asymmetric heterostructures.** Near-field amplitude image in a twisted heterostructure formed by a thin $\alpha$-MoO$_3$ layer with thicknesses $d_{top}$ = 200 nm (**a**, **c**) and 400 nm (**b**) on top of a 500-$\mu$m thick (010) $\beta$-Ga$_2$O$_3$ substrate with twist angles $\theta = 45^\circ$ (**a**), $0^\circ$ (**b**), and $100^\circ$ (**c**) at an illuminating frequency of $\omega = 734$ cm$^{-1}$. A 1-$\mu$m diameter hole allows for the effective launching of the PhPs, whose propagation is visualized by s-SNOM (scattering-type scanning near-field optical microscopy). The experimental iso-frequency curves, obtained by performing the Fourier Transforms (2D-FFT) of the near-field image, are shown in the insets. **d**–**f** Simulated near-field amplitude images of the system in **a** (**d**), **b** (**e**), and **c** (**f**). The 2D-FFTs of the simulated images are again shown in the insets. Both experimental and simulated images are aligned with the crystallographic axes of the top $\alpha$-MoO$_3$ layer.

understand the dispersion of the asymmetric homostructure as a coupling between two separate modes: the mode of the top layer modified by the positive substrate permittivity of the bottom layer (which remains constant independent of the bottom layer thickness), and a ray-like mode (corresponding to one of the asymptotes in momentum space) of the thick bottom layer. A similar argument can be made to explain the "pinwheel" pattern observed for a twist angle of 60$^\circ$. In this case, one part of the bilayer IFC (exhibiting a canalized behavior, Supplementary Fig. 5) couples with one asymptote of the thick layer to generate the observed propagation.

**Unidirectional ray polaritons in twisted heterostructures**

To experimentally demonstrate URPs also in the case of a heterostructure, we fabricated stacks of thin $\alpha$-MoO$_3$ layers over a thick (010) $\beta$-Ga$_2$O$_3$ substrate ($d_{bot}$ = 500 $\mu$m) at three different twist angles $\theta = 0^\circ$, 45$^\circ$ and 100$^\circ$ (see Methods). These angles were estimated by using polarization-resolved Raman spectroscopy (Supplementary Fig. 7). A 1-$\mu$m diameter hole was made on the top layer (see Methods) to efficiently launch PhPs, allowing us to directly image their propagation using s-SNOM. The lower-energy optical phonon resonances of $\beta$-Ga$_2$O$_3$ require longer wavelength excitation not readily available from commercially available table-top sources. Instead, a broadly tunable infrared free-electron laser[42–45] (IR-FEL) coupled to an s-SNOM was employed (see Methods) for near-field imaging of ray polaritons in the twisted heterostructure.

The experimental images for three different twist angles obtained at an illumination frequency of $\omega = 734$ cm$^{-1}$ are shown in Fig. 4a–c. A cross-like pattern with the same orientation and open angle appears for the three twist angles. However, the propagation lengths of each ray differ between the three structures. For a twist angle of 0$^\circ$ (Fig. 4b), both rays exhibit a similar behavior. In contrast, for $\theta = 45^\circ$ and 100$^\circ$ (Fig. 4a, c, respectively) there is a notable asymmetry between them. For $\theta = 45^\circ$ (Fig. 4a), one ray extends much further than the other (and further than in the more symmetric case shown in Fig. 4b), although both rays can still be clearly resolved. For $\theta = 100^\circ$ (Fig. 4c), the directions are flipped, and the other ray shows much longer

propagation. The asymmetry is now more pronounced, with the short ray almost completely suppressed. These data provide clear evidence that this asymmetric heterostructure can support URPs. The asymmetric intensity between the rays of each twist angle is strongly supported by the asymmetric intensity distribution of the cross-like shape observed in the momentum-space representation (insets in Fig. 4a–c) obtained by performing the 2D-FFT of the experimental images. Notably, and in contrast to the homostructure, the twist-induced asymmetry for the heterostructure modulates not only the intensity but also the propagation lengths of the rays (Supplementary Fig. 15), with values ranging from 1$\mu$m for the most suppressed rays to 5$\mu$m for the most enhanced ones. This finding underlines a different physical origin of unidirectional ray formation in the two systems.

To corroborate the experimental results, we again performed full-field numerical simulations, as depicted in Fig. 4d–f (see Methods for details), which show a remarkable agreement with the experimental real-space images. Note that the shape, the direction of propagation, and the asymmetry between the rays align well with the experimental data. Interestingly, the numerical simulations, in contrast to the experimental images, show hyperbolic-like propagation along the [001] $\alpha$-MoO$_3$ axis with a considerably reduced wavelength ($\sim$ 300nm). Waves of such small wavelengths, associated with high-order modes of the system, cannot be efficiently excited by the 1-$\mu$m hole in the experiments. Consequently, only ray-like PhPs signatures appear in the experimental data. A similar behavior occurs for different illumination frequencies (Supplementary Fig. 8). However, the asymmetry between the rays varies with frequency for a given twist angle, mainly due to the frequency-dependent rotation of the $\beta$-Ga$_2$O$_3$ optical axis associated with the shear effect recently reported in this monoclinic crystal[24–26].

The direction of propagation of ray-like PhPs in the heterostructure remains unchanged independent of the $\beta$-Ga$_2$O$_3$ twist angle, as observed in Fig. 4. In fact, this direction coincides with the asymptotes for PhPs of the $\alpha$-MoO$_3$ top layer, which are defined by Eq. (1). For $\omega = 734$ cm$^{-1}$ this equation gives rise to an in-plane angle of $\pm 61^\circ$ with respect to the [100] crystal direction of the top $\alpha$-MoO$_3$ layer, in the

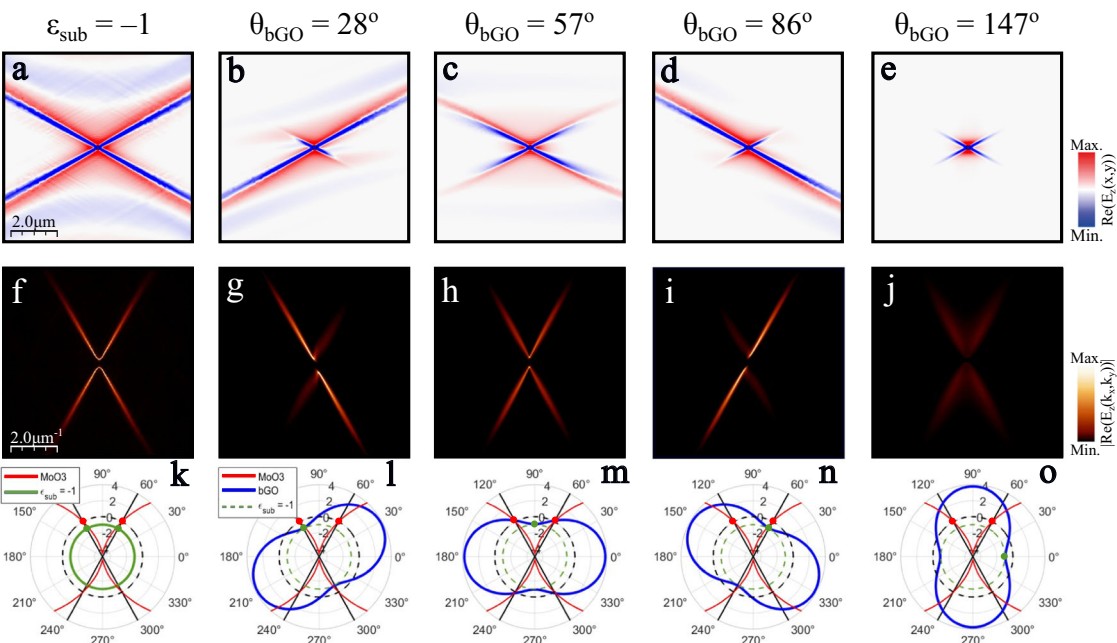

**Fig. 5 | Analysis of the ray asymmetry in twisted asymmetric heterostructures.** **a**–**e** Simulated near-field amplitude images of a system made of a 100-nm thin $\alpha$-MoO$_3$ layer over an isotropic substrate with permittivity $\varepsilon_{sub} = -1$ (**a**), and over a 5-μm thick $\beta$-Ga$_2$O$_3$ substrate with twist angles $\theta = 28^o$ (**b**), $57^o$ (**c**), $86^o$ (**d**) and $147^o$ (**e**). The illumination frequency is $\omega = 734$ cm$^{-1}$. **f**–**j** Simulated isofrequency curves of **a**–**e** obtained by performing the Fourier Transforms (2D-FFTs) of the simulated near-field images **a**–**e**, respectively. **k**–**o**. In-plane real permittivity values of $\alpha$-MoO$_3$ (red curve), an isotropic material with permittivity $\varepsilon_{sub} = -1$ (green curve) and $\beta$-Ga$_2$O$_3$ (blue curve). The black straight lines represent the two angular directions of the $\alpha$-MoO$_3$ asymptotes which are defined by a zero-permittivity $\alpha$-MoO$_3$ value (red spots). The permittivity value of $-1$ for $\beta$-Ga$_2$O$_3$ is marked by a green spot. When the green and red spots are aligned, unidirectional ray-like propagation occurs along the corresponding asymptote. The black (green) dashed curve corresponds to the value $\varepsilon = 0$ ($\varepsilon = -1$) and is present for visual guidance.

momentum space representation, corresponding to a real space propagation direction of $\mp 29^o$. Although these angles align well with the direction of propagation observed in the experimental images, high losses are expected for PhPs whose propagation is dominated by the IFC asymptotes of the $\alpha$-MoO$_3$ layer. Consequently, the $\beta$-Ga$_2$O$_3$ substrate must be the main reason why these asymptotic modes appear while it simultaneously also introduces asymmetry to the PhP dispersion.

Interestingly, a symmetric ray-like propagation effect has also been observed in a single thin $\alpha$-MoO$_3$ layer over a SiC substrate[22] at an illumination frequency of $\omega = 943$ cm$^{-1}$. This frequency corresponds to the surface dipole excitation of SiC where the real part of the dielectric permittivity takes the value $-1$[46]. At this condition, two PhP rays propagate along the asymptotic $\alpha$-MoO$_3$ directions. To corroborate whether a similar effect is happening for our heterostructure, we first simulate the PhP propagation in a thin $\alpha$-MoO$_3$ on top of an isotropic substrate with a permittivity value of $\varepsilon_{sub} = -1$ at an illumination frequency of $\omega = 734$ cm$^{-1}$ (Fig. 5a). As in ref. 22, a cross-like pattern arises for this artificial system. In this context, it is useful to analyze the permittivity of the $\beta$-Ga$_2$O$_3$ substrate at this frequency. Indeed, $\beta$-Ga$_2$O$_3$ supports in-plane hyperbolic PhPs at $\omega = 734$ cm$^{-1}$, with the real part of the projected permittivity $\varepsilon_{bGO}^{\varphi}$ ranging from $4\mathrm{Re}\left(\varepsilon_{bGO}^{\varphi}\right) - 1$. In particular, there is a single in-plane angle $\varphi_c$ for which a similar condition ($\varepsilon_{sub} = -1$) appears. Specifically, $\varepsilon_{bGO}^{\varphi_c} = -0.94 + 0.11i$ and $\varepsilon_{bGO}^{z} = -0.95 + 0.03i$ for $\varphi_c = 33^o$ with respect to the $\beta$-Ga$_2$O$_3$ a-axis direction. Thus, a unique condition arises when this specific direction of the $\beta$-Ga$_2$O$_3$ substrate aligns with the $\alpha$-MoO$_3$ asymptotes at certain twist angles, which is largely responsible for the formation of the URPs that we observe.

To corroborate this interpretation, we calculated the near-field response of the heterostructure at several twist angles $\theta = 28^o$, $57^o$, $86^o$, and $147^o$, see Fig. 5b–e, respectively, along with the respective FFTs shown in Fig. 5g–j. By choosing these specific values, we align the

unique direction $\varphi_c$ of the $\beta$-Ga$_2$O$_3$ substrate from being along and in between the $\alpha$-MoO$_3$ asymptotes. Clearly, extreme asymmetry with almost entirely unidirectional ray-like propagation is achieved if $\varphi_c$ is aligned with either asymptote of the $\alpha$-MoO$_3$ layer, while symmetric rays emerge for $\varphi_c$ aligned between the asymptotes. Notably, the unidirectional rays propagate significantly further than the symmetric ones, further corroborating the enhancement of ray-like propagation at the approximate $\varepsilon_{sub} = -1$ condition. The relative alignment of the projected permittivities of both materials for each twist angle is detailed in Fig. 5l–o. Additionally, the green dots mark the $\varphi_c$ direction of the $\beta$-Ga$_2$O$_3$ substrate, and the red dots mark the direction of the asymptotes of the $\alpha$-MoO$_3$ layer, graphically illustrating the alignment of the critical directions of both materials in the twisted heterostructure. For the anisotropic cases, Fig. 5l,n, the suppressed rays emerge at directions where the substrate projected permittivity is positive ($\varepsilon_{bGO}^{\varphi} = 2.5 + 1.5i$ and $\varepsilon_{bGO}^{\varphi} = 2.5 + 2i$ for Fig. 5l, n, respectively), which is responsible for suppressing the propagation. Note that the imaginary part of these values is different due to the shear effect recently demonstrated in this monoclinic material[24–26]. This suppression of either ray for the asymmetric patterns is consistent with strong suppression of both rays at $\theta = 147^o$ (Fig. 5e, j, o) where the substrate projected permittivity is positive and large along both asymptotes, while at $\theta = 57^o$ (Fig. 5c, h, m) the substrate permittivity takes values near zero along the asymptotes resulting in intermediate propagation lengths for both rays. The gradual suppression of polariton propagation for directions away from $\varphi_c$ condition is related not only to the real part but also to the imaginary part of the projected substrate permittivity (see Supplementary Figs. 12–14 for details), modifying not only confinement but also optical losses for the polaritons through twisting the heterostructure layers. Thus, the $\beta$-Ga$_2$O$_3$ substrate offers an anisotropic permittivity able to support the asymptotic polariton propagation in the $\alpha$-MoO$_3$ layer within a narrow angular region. By twisting the heterostructure we can select the

degree of asymmetry between the two $\alpha$-MoO$_3$ asymptotes, allowing for the generation of URPs whose direction is locked to the asymptotes of the $\alpha$-MoO$_3$ layer.

Our analysis shows that the physical reason responsible for the appearance of URPs is the alignment of the negative $\beta$-Ga$_2$O$_3$ radial permittivity values with the $\alpha$-MoO$_3$ asymptotes. In contrast, the shear effect associated with the monoclinic $\beta$-Ga$_2$O$_3$ does not play a major role for the unidirectional ray formation. However, there are some features that cannot be understood without considering this shear effect. (i) The slight asymmetry between the two rays in the symmetric cases (Fig. 5c, h, m and Fig. 5e, j, o) is caused by the asymmetric imaginary response of $\beta$-Ga$_2$O$_3$ (Supplementary Figs. 12–14). (ii) The optimal twist angle for which unidirectional ray-like PhPs arise varies with frequency. Apart from the frequency dependence of the $\alpha$-MoO$_3$ PhP asymptote directions, also the optical axis direction in $\beta$-Ga$_2$O$_3$ changes with frequency, leading to a nontrivial evolution of ray-like PhPs with frequency (Supplementary Fig. 11).

## Discussion

The two asymmetric structures considered in this work have been found to both support URPs with a high degree of tunability and unique properties. On the one hand, the direction of URPs in the homostructure, formed by two $\alpha$-MoO$_3$ layers, can be modified by simply varying the excitation frequency. On the other hand, in the twisted heterostructure formed by a thin $\alpha$-MoO$_3$ layer and a thick $\beta$-Ga$_2$O$_3$ substrate, the asymmetry of the two rays can be tuned by means of both twist angle and illumination frequency variations. We explain the PhP response in the homostructure as the result of the coupling of two separate modes: a mode of the top thin $\alpha$-MoO$_3$ layer modified by the positive bottom layer permittivity and a ray-like mode of the bottom thick $\alpha$-MoO$_3$ layer. This is the reason why modifying the twist angle changes dramatically the PhP response, i.e., from unidirectional ray-like PhPs to "pinwheel" shaped PhPs. In contrast, the URPs in the heterostructure follow a different mechanism. At a given frequency, the narrow angular sector of negative projected permittivity of the $\beta$-Ga$_2$O$_3$ substrate enhances PhP rays if twist-aligned with the asymptotes of the $\alpha$-MoO$_3$ layer. By twisting the structure, we can control the degree of asymmetry between the two rays. The monoclinic nature of $\beta$-Ga$_2$O$_3$ and the associated shear effects play a minor role, and the polariton features emerge solely from its in-plane anisotropy. Notably, the different mechanisms for URP formation between the homostructure and the heterostructure result in different twist-induced effects on the propagation lengths for each case, as detailed in Supplementary Figs. 6 and 15. Interestingly, the URPs observed in this work are characterized by a propagation at constant phase along a direction of effective zero permittivity suggesting similarities with epsilon-near-zero physics[47–49], such as a constant phase, diffraction-less propagation, yet with extreme intrinsic directionality, leading to a large range of open fundamental questions, as well as unique opportunities for photonic applications that rely on nanoscale waveguiding and routing.

In conclusion, by employing asymmetrically stacked, twisted biaxial crystals, we demonstrate the natural emergence of unidirectional ray PhPs. We experimentally visualize these polaritonic excitations in two twisted asymmetric structures involving either a thin and a thick layer of the orthorhombic crystal $\alpha$-MoO$_3$ or one thin layer of $\alpha$-MoO$_3$ and a thick substrate of the monoclinic crystal $\beta$-Ga$_2$O$_3$. We find that the URPs supported in these structures can be tuned by means of variations in the twist angle and the illumination frequency. These features are crucial for the implementation of twisted asymmetric structures in optical nanotechnologies. In addition, exotic PhP propagations, such as pinwheel PhPs, arise naturally from these structures. We theoretically explain the appearance of unidirectional ray propagation in the homo- and heterostructure systems, providing much insight into the complex excitation and propagation of PhPs in

twisted systems. In this regard, the large variety of hyperbolic vdW materials, as well as hyperbolic 3D-crystals, opens a plethora of possibilities for exploring the limits of PhP propagation in twisted multi-layers and provides a large material base for nanophotonic applications.

During the revision of this manuscript, we became aware of a theoretical work studying the appearance of lateral optical forces generated by polaritons in asymmetric stacks of twisted $\alpha$-MoO$_3$ layers with a configuration similar to our homostructure[50].

## Methods

### Fabrication of twisted stacks

The twisted stacks were fabricated using the dry transfer technique[51]. $\alpha$-MoO$_3$ layers were extracted from mechanical exfoliation of commercial $\alpha$-MoO$_3$ bulk materials (Alfa Aesar) using Nitto tape (Nitto Denko, SPV 224 P). For the homostructure, a thin and a thick $\alpha$-MoO$_3$ flake were exfoliated from the Nitto tape to a transparent poly-(dimethylsiloxane) (PDMS) stamp, where selection of the desired thicknesses was carried out using an optical microscope. Subsequently, we employed a home-made micromanipulator to align and twist the PDMS stamps with the $\alpha$-MoO$_3$ flakes forming the stack. To do this, we first released the thick $\alpha$-MoO$_3$ flake on a SiO$_2$ substrate by heating the PDMS stamp to 200 °C, and finally, we placed the thin $\alpha$-MoO$_3$ flake on top of it while twisting it at the desired angle. For the heterostructure, we used commercially available wafer samples of (010) $\beta$-Ga$_2$O$_3$ doped with Fe to compensate for inherent free carriers ($\sim10^{12}$ cm$^{-3}$). We again use the dry transfer technique (following the steps described before) to place several twisted thin $\alpha$-MoO$_3$ layers on top of the $\beta$-Ga$_2$O$_3$ crystal.

### Fabrication of the nanoholes

For the homostructure, the thin $\alpha$-MoO$_3$ layers were milled through by FIB processing[52] using a FEI Helios 600 Nanolab FIB–SEM system. The optimized parameters to produce holes with an average diameter of $224.9 \pm 7.6$ nm were an ion beam voltage of 30 kV, an ion beam current of 1.5 pA and a dwell time of 2 $\mu$s. The nanoholes of the heterostructure were milled using a FEI Helios NanoLab G3 CX focused ion beam (FIB) with an ion beam voltage of 30 kV and an ion beam current of 7.7 pA. To remove gallium ion intercalation from both systems, we annealed our samples at 300 ℃ for 90 minutes in the ambient air.

### Scattering scanning near-field optical microscopy

For the homostructure, near-field imaging measurements were performed employing a commercial scattering-type scanning near-Field optical microscope (s-SNOM) from Neaspec GmbH, equipped with a quantum cascade laser from Daylight Solutions (890-1140 cm$^{-1}$). Metal-coated (Pt/Ir) atomic force microscopy (AFM) tips (ARROW-NCPt-50, Nanoworld) at a tapping frequency $\Omega \sim 280$ kHz and an oscillation amplitude $\sim 100$ nm were used as source and probe of polaritonic excitations. Both the hole and the AFM tip were illuminated with s-polarized infrared light from the quantum cascade laser. The light scattered by the tip with the near-field information was focused by a parabolic mirror into an infrared detector (Kolmar Technologies) in the far field. Demodulation of the detected signals n$\Omega$, which can be written as the complex-valued functions $\sigma_n = s_n e^{i\phi_n}$, was performed to the 3rd harmonic ($n = 3$) of the tip frequency for background suppression. A pseudo-heterodyne interferometric method was employed to independently extract both amplitude ($s_3$) and phase ($\phi_3$) signals.

For the heterostructure, near-field imaging measurements were performed using a commercial s-SNOM system from Neaspec, coupled to the free-electron laser FELBE at the Helmholtz–Zentrum Dresden–Rossendorf[53]. Experimental details are equivalent to those in previous works[26]. In short, by using the FEL as the infrared light source we rely on self-homodyne measurements rather than pseudo-

heterodyne due to the relative instability of the FEL. This results in intermixed amplitude and phase signals.

## Full-wave numerical simulations

The full-wave numerical simulations were performed using the software COMSOL Multiphysics, based on the finite boundary elements method. The structure was composed of 2 semi-infinite media (superstrate and substrate) with a thin and a thick layer in between and a vertically oriented electric dipole on top of the flake acting as a polaritonic launcher. We calculate the vertical component of the electric field $Re(E_z)$ at 5 nm on top of the uppermost surface of the twisted stacks, which is then subjected to 2D-FFT to extract the simulated PhP IFCs. Momentum axis of the IFCs is defined as $k = \frac{1}{\lambda}$. The dielectric permittivity values for $\alpha$-$MoO_3$ (ref. 18) and $\beta$-$Ga_2O_3$ (ref. 24) are taken from elsewhere.

## Data availability

All data that support the findings of this study are present in the paper and the Supplementary Information. All raw data generated during the current study are available from the corresponding authors upon request.

## Code availability

All code used in this study is available from the corresponding author upon request.

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

## Acknowledgements

J.A.-C., A.I.F.T.-M., E.T.-G. and G.A.-P. acknowledge support from the Severo Ochoa program of the government of the Principality of Asturias (nos. PA-22-PF-BP21-100, PA-21-PF-BP20-117, PA-23-PF-BP22-046 and PA-20-PF-BP19-053, respectively). J.M.-S. acknowledges financial support from the Ramón y Cajal Program of the Government of Spain and FSE (RYC2018-026196-I), and the project PCI2022-132953 funded by MCIN/AEI/10.13039/501100011033 and the EU "NextGenerationEU/PRTR". P.A.-G. acknowledges support from the European Research Council under Consolidator grant no. 101044461, TWISTOPTICS and the Spanish Ministry of Science and Innovation (State Plan for Scientific and Technical Research and Innovation grant number PID2022-141304NB-I00). A.Y.N. acknowledges the Spanish Ministry of Science and Innovation (grants PID2020-115221GB-C42, PID2023-147676NB-I00) and the Basque Department of Education (grant PIBA-2023-1-0007). L.H. and J.M.D.T. acknowledge financial support by Gobierno de Aragón via grant E13_23R with European Social Funds (Construyendo Europa desde Aragón). L.H. and J.M.D.T. thank the CSIC Interdisciplinary Thematic Platform (PTI) on Quantum Technologies (PTI-QTEP). Authors acknowledge the use of instrumentation as well as the technical advice provided by the National Facility ELECMI ICTS, node «Laboratorio de Microscopias Avanzadas (LMA)» at «Universidad de Zaragoza». Parts of this research were carried out at the ELBE Center for High-Power Radiation Sources at the Helmholtz–Zentrum Dresden–Rossendorf e. V., a member of the Helmholtz Association. M.O., L.M.E., and S.C.K. acknowledge the financial support by the Bundesministerium für Bildung und Forschung (BMBF, Federal Ministry of Education and Research, Germany, Project Grant Nos. 05K19ODB and 05K22ODA) and by the Deutsche Forschungsgemeinschaft (DFG, German Research Foundation) under Germany's Excellence Strategy through Würzburg-Dresden Cluster of Excellence on Complexity and Topology in Quantum Matter—ct.qmat (EXC 2147, project-id 390858490). T.G.F. was supported through the National Science Foundation through grant 2236807. S.D. was supported by the Office of Naval Research MURI grant N0001-23-1-2567. K.D.-G. acknowledges support by the Army Research Office under grant W911NF-21-1-0119. R.K. acknowledges support by the National Aeronautics and Space Administration (NASA) grant NASA 80NSSC22K1201. A.S.S. acknowledges support by the Office of Naval Research Petersen grant N00014-23-1-2676 while J.D.C. was supported by the Office of Naval Research grant N00014-22-1-2035 and by the Department of Energy—Basic Energy Sciences under Grant DE-FG02-09ER4655. G.C., N.S.M., S.W., M.W., and A.P. were supported by the Max Planck Society.

## Author contributions

J.A.-C., M.O., and S.D. contributed equally to this work. A.I.F.T.-M., E.T.-G. and A.T.M.-L. prepared the twisted $\alpha$-MoO$_3$ samples. S.D., A.I.F.T.-M., L.H., and J.M.T. performed the focussed-ion-milling of holes into the $\alpha$-MoO$_3$ flakes. T.B. performed Raman measurements to determine the twist angles. J.A.-C., A.I.F.T.-M., E.T.-G., and P.A.-G. carried out the near-field measurements of the $\alpha$-MoO$_3$ stacks. M.O., S.D., G.C., G.A.-P., K.D.G., R.K., A.S.S., N.S.M., S.W., J.M.K., T.G.F., J.D.C., and A.P. performed the near-field measurements of the $\alpha$-MoO$_3$/$\beta$-Ga$_2$O$_3$ stacks. J.A.-C, S.D., C.L., L.F.-A., and A.P. carried out the numerical calculations. M.W., S.C.K., L.M.E, J.M.-S., A.Y.N., J.D.C, and P.A.-G. acquired part of the funding required to carry out this work. J.A.-C., P.A.-G., and A.P. wrote a first version of the manuscript that was edited by the rest of the authors. S.C.K., A.Y.N., J.D.C., P.A.-G., and A.P. supervised the work. J.A.-C., P.A.-G., and A.P. conceived the original idea.

## Funding

## Competing interests

The authors declare no competing interests.
