## [Peer Review File · Nature Communications]

Unidirectional Ray Polaritons in Twisted Asymmetric StacksREVIEWER COMMENTS

Reviewer #1 (Remarks to the Author):

The paper “Unidirectional Ray Polaritons in Twisted Asymmetric Stacks” is a collection of state-of-the-art measurements on different Layered Materials stacks, also involving the use of a broadband FEL source. I find it rigorous in both experiments and theory. However, the reported results do not match, to my opinion, the advancement and impact, with respect to previous literature, to be considered for publication in Nature Communications.

The literature about addressing and manipulating phonon polaritons in layered materials has exploded in the last few years. Unidirectional ray polaritons have been already reported several times, also by some of the authors of this paper. Although the authors propose here yet another combination of materials, the resulting ray-like behavior is well-known and can be achieved with other configurations. I personally also do not agree with the tentative of justify the importance of the presented configuration by pointing at the tunability of the PhPs with frequency. If any of these effects are ever going to be used practically, the potential user would prefer to choose the propagation angle at a design wavelength rather than be forced to use a specific wavelength to get that propagation angle.

In the contest of the presented results, I find the only “unprecedented” observation in the “pinwheel” pattern. However, also in this case, I find the impact of such realization quite low. In fact, the authors also introduce the related theory as justification of a pattern found during near-field imaging, rather than an engineered condition that would advance the use or impact of PhPs in layered materials.

Based on the comments above, I would not support publication of this work in Nature Communications.

Reviewer #2 (Remarks to the Author):

The manuscript authored by J. Álvarez-Cuervo presents detailed observations and innovative material design strategies for achieving Unidirectional Ray Polaritons within a twisted anisotropic polaritonic medium. The authors have initially established, both conceptually and numerically, the presence of unidirectional ray polaritons in twisted MoO₃ flakes characterized by asymmetric layer thickness. Subsequently, the proposed unidirectional ray polaritons were experimentally validated in both the twisted MoO₃/MoO₃ and MoO₃/Ga₂O₃ bilayer systems, encompassing a range of thicknesses. The experimental data is presented with clarity and is well-supported by comprehensive numerical simulation. Given by the robust experimental evidence and the solid numerical/theoretical backing, the proposed unidirectional ray polaritons stand as a promising platform for achieving unidirectional light propagation across broad frequency ranges with

considerable tunability. Nevertheless, there are a few minor aspects that, if addressed, could significantly enhance the manuscript's readability and scientific integrity. It would endorse its publication in Nature Communications, conditional upon the authors making the revision based on the following point:

1. Line 40 "the presence of a single phase of the propagating field" : Given that a propagating electromagnetic wave typically exhibits an oscillating phase due to the absence of net charge, it would be more precise to describe this characteristic in terms of its behavior along the propagation direction. Therefore, I recommend rephrasing the sentence.
2. Line 98 mentions "a strong symmetry breaking", which lacks a clear definition within the manuscript. Considering that the presence of a twisted angle inherently disrupts either inversion or mirror symmetry, it is essential to specify the nature of the symmetry breaking observed. A more detailed explanation of the disparity's role will enhance the reader's comprehension of the material's symmetrical properties.
3. Line 125-128 describe a variation in the field intensity of rays within the simulation, noting an increase for one ray and a decrease for another as the twisted angle is altered. However, the description does not specify whether these changes refer to the absolute intensity or the relative intensity in relation to the twisted angle. The interpretation of these changes could also be influenced by the definition of the colormap used in the simulation. To avoid ambiguity, I recommend that the author provides a more detailed explanation, clearly distinguishing between absolute and relative intensity changes.
4. Line 177-178 refer to "a bright polaritonic fringe that propagates." However, the term "fringe" typically denotes conventional polaritonic patterns that are parallel to the edge of the material and perpendicular to the direction of propagation. To prevent any potential misinterpretation, I recommend substituting "fringe" with a term that more accurately reflects the observed phenomenon.
5. Line 220-221 states that the reduction of intensities and the increase of losses enable Unusual reflection patterns. However, the current presentation does not make it evident that these two effects indeed occur concurrently. To substantiate this claim, I recommend that the author conducts a more thorough analysis of the data. It would be beneficial to include a quantitative discussion or a graphical representation that clearly demonstrates the correlation between intensity reduction and loss increase. Providing more solid evidence will strengthen the argument and enhance the credibility of the findings.
6. Line 284-282 provide an interpretation of the physical origins of unidirectional propagation, which I find to be a pivotal aspect of the study. To further strengthen this key point, I suggest that the authors include visualizations from 3D COMSOL simulations . Specifically, resending the vertical cross-section of the electric field profile along the radial direction at various angles would greatly enhance the reader's understanding of the phenomenon.
7. Line 327 addresses the propagation lengths of the rays, yet the term is not adequately defined within the manuscript. The concept of propagation length for ray polariton may be complex, as it would encompass decay due to material loss or the dispersion of electromagnetic energy across a finite energy. To provide clarity, I recommend that the author offers a comprehensive definition of the propagation length specific to Ray polaritons. Additionally, detailing the extraction process used to determine this value would greatly benefit the reader's understanding.

Reviewer #3 (Remarks to the Author):

The authors reported on the unidirectional propagation of phonon polaritons in van der Waals stacks composed of homo- and heterostructures. They demonstrated that by forming homostructures of α -MoO₃ and heterostructures of α -MoO₃ and β -Ga₂O₃, the unidirectional flow of phonon polaritons can be realized by either tuning the excitation frequency or controlling the tuning angles. In stark contrast to previous studies on twisted bilayer α -MoO₃, where the thicknesses of the two α -MoO₃ layers are very thin and comparable to each other, the steering of phonon polaritons in the current study is achieved in stacks with very different thicknesses of their constituents. The underlying mechanisms governing such steering are discussed and show good agreement with the experimental findings.

Steering of phonon polaritons in the basal plane of van der Waals polar crystals is crucial for their applications in subwavelength waveguiding. The concept of twisted optics, which stacks two van der Waals slabs perpendicularly with the optical axes mismatched between the top and bottom constituents, has already proven to be an efficient approach. In contrast to previous studies, the authors proposed a more facile method to construct the twisted structure, requiring only one thin van der Waals layer as the top layer. In this way, the steering of polaritons occurs due to the modulation of the polariton propagation either by the dielectric environment provided by the thick bottom layer or by the hybridization of polaritons from both layers, depending on the excitation frequency employed. I find the results impressive and believe they are suitable for publication in Nature Communications. Here are a few concerns on the scientific points that may help improve the presentation of their study.

- 1) The most significant innovation of this work is the use of vertically stacked van der Waals crystals with different thicknesses to achieve unidirectional propagation of phonon polaritons. The authors have indeed observed this phenomenon experimentally. However, they need to discuss the transformative breakthroughs this method could bring to practical applications, especially in comparison to previously reported results.
- 2) In Figure 2 and 4, it is suggested to provide the dependence of unidirectional flow direction on the stacking angle (with fixed excitation frequency) and excitation frequency (with fixed stacking angle).
- 3) The thickness of the top layer is indeed much thinner than that of the bottom layer. I would like to know what would happen if the thickness of the top layer is reduced to a few atomic layers or even a single atomic layer.
- 4) In the discussion above Figure 2, the authors mention that the propagation direction of phonon polaritons can be actively controlled. However, the current experiments are actually based on specific frequencies and structures, and do not achieve reversible control. It is recommended to temper such description.
- 5) In discussion above Figure 1, the authors state that “the other branch is suppressed completely”.

This is not entirely accurate, as there is still energy flow along the weaker branch. The authors may consider modifying this description.

Response to the reviewers comments:

Reviewer #1 (Remarks to the Author):

The paper "Unidirectional Ray Polaritons in Twisted Asymmetric Stacks" is a collection of state-of-the-art measurements on different Layered Materials stacks, also involving the use of a broadband FEL source. I find it rigorous in both experiments and theory. However, the reported results do not match, to my opinion, the advancement and impact, with respect to previous literature, to be considered for publication in Nature Communications.

The literature about addressing and manipulating phonon polaritons in layered materials has exploded in the last few years. Unidirectional ray polaritons have been already reported several times, also by some of the authors of this paper. Although the authors propose here yet another combination of materials, the resulting ray-like behavior is well-known and can be achieved with other configurations. I personally also do not agree with the tentative of justify the importance of the presented configuration by pointing at the tunability of the PhPs with frequency. If any of these effects are ever going to be used practically, the potential user would prefer to choose the propagation angle at a design wavelength rather than be forced to use a specific wavelength to get that propagation angle.

In the contest of the presented results, I find the only "unprecedented" observation in the "pinwheel" pattern. However, also in this case, I find the impact of such realization quite low. In fact, the authors also introduce the related theory as justification of a pattern found during near-field imaging, rather than an engineered condition that would advance the use or impact of PhPs in layered materials.

Based on the comments above, I would not support publication of this work in Nature Communications.

We thank the reviewer for finding our work rigorous, both experimentally and theoretically. However, we respectfully disagree on multiple accounts with the assessment of our work in terms of impact, as detailed in the following:

First, the reviewer states that unidirectional ray polaritons have already been reported several times. As we also discuss in the manuscript, the difference with previous publications is that in our work, the unidirectional ray character emerges from the intrinsic properties of the material stack, whereas previous reports (ref. 26,27) require specifically designed antennas and polarization conditions to suppress certain components of the polariton dispersion. Therefore, our work is both fundamentally different as well as of more practical use since the unidirectionality will persist as an intrinsic feature even if the antenna structure needs to be adapted to match the demands of a specific application. The ray-like behaviour supported in a natural system, to the best of our knowledge, has only been reported once (ref. 22), but being bi-directional instead of uni-directional, for a specific frequency and without any tunability of its intrinsic properties. In that regard, the reviewer's statement "... resulting ray-like behavior is well-known and can be achieved with other configurations ..." is somewhat unclear. We would kindly ask the reviewer to be more specific to allow for a proper response.

Second, the reviewer questions the advantage of frequency tuning of the ray direction, providing one "typical" design demand. We would argue that, in fact, one can very well imagine applications where the frequency-dependent polariton direction would be very beneficial. Imagine, for instance, a sensing device that would distinguish analytes with closely spaced resonances. Here, the frequency-dependent routing direction would allow a nanoscale on-chip spatial separation of the selective information using a broadband source. We note, additionally, that designing a specific direction for a specific wavelength is still perfectly feasible due to the in-plane anisotropic nature of the employed materials, such that rotating the crystal is always an option to guide the waves in a specific direction. Additionally, as highlighted by the reviewer 3 and now remarked in the main manuscript, to fabricate our stack only one thin vdW layer is required, which constitutes a significant simplification with respect to previous reports of unidirectional PhP propagation where double or triple thin layers were used.

Changes in the revised manuscript, line 97:

"Consequently, only one thin vdW layer is required to fabricate these stacks which constitutes a significant simplification with respect to previous reports of unidirectional PhP propagation where double or triple thin layers were used."

Finally, the reviewer questions the relevance of the observation of the pinwheel pattern. While obviously not being the focus of the work, we believe that reporting that "accidental finding" (the reviewer is correct in realizing that this was not an engineered case) is highly relevant. It demonstrates the large variety of different possible polariton propagation features and, thus, opens up many opportunities for nanoscale design of engineered light propagation similar to optical metasurfaces, yet employing polaritons in highly anisotropic media. We believe this field may very well develop much further than expected by the reviewer, and we would therefore argue that this is indeed a relevant finding.

Overall, we would argue that our work provides novelty both in terms of fundamental insight as well as the potential for applications to warrant publication in Nature Communications, as additionally highlighted in the revised abstract.

Reviewer #2 (Remarks to the Author):

The manuscript authored by J. Álvarez-Cuervo presents detailed observations and innovative material design strategies for achieving Unidirectional Ray Polaritons within a twisted anisotropic polaritonic medium. The authors have initially established, both conceptually and numerically, the presence of unidirectional ray polaritons in twisted MoO₃ flakes characterized by asymmetric layer thickness. Subsequently, the proposed unidirectional ray polaritons were experimentally validated in both the twisted MoO₃/MoO₃ and MoO₃/Ga₂O₃ bilayer systems, encompassing a range of thicknesses. The experimental data is presented with clarity and is well-supported by comprehensive numerical simulation. Given by the robust experimental evidence and the solid numerical/theoretical backing, the proposed unidirectional ray polaritons stand as a promising platform for achieving **unidirectional light propagation across broad frequency ranges with considerable tunability**. Nevertheless, there are a few minor aspects that, if addressed, could significantly enhance the manuscript's readability and scientific integrity. It would endorse its publication in Nature Communications, conditional upon the authors making the revision based on the following point:

We thank the reviewer for the insightful assessment of our work and the appreciation of our results regarding the importance, novelty, and well-supported presentation. In the following, we address point by point the raised comments and questions.

1. Line 40 "the presence of a single phase of the propagating field" : Given that a propagating electromagnetic wave typically exhibits an oscillating phase due to the absence of net charge, it would be more precise to describe this characteristic in terms of its behavior along the propagation direction. Therefore, I recommend rephrasing the sentence.

We thank the reviewer for this comment. Actually, the reviewer is completely right and the phrase is a bit misleading. Rays are characterized by the perpendicular nature between the momentum k and the pointing vector S . Thus, as pointed out by the reviewer, along the direction of propagation, no fringes are expected, resulting in a unique phase at a given time. To be more specific, we rewrite this phrase as follows:

Previous text, line 40:

"URPs are characterized by the absence of diffraction and the presence of a single phase of the propagating field."

Revised text, line 38:

"Importantly, their ray-like propagation, characterized by large momenta and constant phase, is tunable by both the twist angle and the illumination frequency."

Please note that we rewrote the abstract overall, to sharpen the message but also to confide with the journal length requirements.

2. Line 98 mentions "a strong symmetry breaking", which lacks a clear definition within the manuscript. Considering that the presence of a twisted angle inherently disrupts either inversion or mirror symmetry, it is essential to specify the nature of the symmetry breaking observed. A more detailed explanation of the disparity's role will enhance the reader's comprehension of the material's symmetrical properties.

We thank the reviewer very much for pointing out this issue. Indeed, polariton wavelengths are strongly dependent on the layer thicknesses. In a bilayer composed of layers with similar thicknesses, the polariton momenta in the two layers naturally match, therefore allowing for their interaction. For bilayers composed of layers featuring different thicknesses, the polariton momenta are not similar, which can lead to intriguing new shapes of the propagating wavefronts, as reported in our manuscript. However, as pointed out by the reviewer, there is no specific symmetry broken within the system. To be more accurate and considering that wavelength dependence with the layer thicknesses is fully discussed along the text, we reword this section as:

"Here, we report the experimental observation of URPs being naturally supported in twisted asymmetric structures composed of a thin and a thick hyperbolic layer."

3. Line 125-128 describe a variation in the field intensity of rays within the simulation, noting an increase for one ray and a decrease for another as the twisted angle is altered. However, the description does not specify whether these changes refer to the absolute intensity or the relative intensity in relation to the twisted angle. The interpretation of these changes could also be influenced by the definition of the colormap used in the simulation. To avoid ambiguity, I recommend that the author provides a more detailed explanation, clearly distinguishing between absolute and relative intensity changes.

We thank the reviewer for this comment. Indeed, the previous version of the manuscript was somewhat unclear about these magnitudes. Every simulated real space map in Fig. 1c-e, as well as in Fig. 1f-g, uses the same scale such that a direct comparison of increase or decrease of intensity is indeed valid. To avoid any misunderstanding, we now clarify the scales in the revised figure caption. Additionally, we now include a quantitative analysis of the twist-dependent ray amplitudes and propagation lengths in response to the reviewer's question nr. 5, which will further clarify these matters.

4. Line 177-178 refer to "a bright polaritonic fringe that propagates." However, the term "fringe" typically denotes conventional polaritonic patterns that are parallel to the edge of the material and perpendicular to the direction of propagation. To prevent any potential misinterpretation, I recommend substituting "fringe" with a term that more accurately reflects the observed phenomenon.

Indeed, the phrasing in this sentence was a bit unfortunate and potentially misleading. We thank the reviewer for pointing it out. We rephrased it by replacing "fringe" by "ray" as:

"The image obtained for the case of $\theta = 30^\circ$ and $\omega = 880 \text{ cm}^{-1}$ (Fig.2a) shows a bright polariton ray that propagates away from the hole with decaying amplitude along the in-plane direction defined by $\varphi = 30^\circ$ "

5. Line 220-221 states that the reduction of intensities and the increase of losses enable unusual reflection patterns. However, the current presentation does not make it evident that these two effects indeed occur concurrently. To substantiate this claim, I recommend that the author conducts a more thorough analysis of the data. It would be beneficial to include a quantitative discussion or a graphical representation that clearly demonstrates the correlation between intensity reduction and loss increase. Providing more solid evidence will strengthen the argument and enhance the credibility of the findings.

We are extremely grateful for this comment from the reviewer, which forced us to finally look into this question rigorously. We had indeed not analysed in detail whether the suppression of one of the rays is an effect of reduced intensity through reduced launching efficiency from the source or an effect of propagation losses leading to a more rapid decay of the ray. The real space images do not usually clearly show the difference between these two scenarios.

Following the suggestion by the reviewer, we have now analysed the intensities and propagation lengths using our simulations. Notably, extracting the propagation length from the experiment is very difficult for these types of modes (in contrast to more conventional phonon polaritons) since they do not show oscillations along the propagation direction, and it is thus, in practice, very hard to separate (with confidence) the polariton signal from any background drift in the image. We have thus refrained from such analysis for the experimental data. In the simulations, however, this is straightforward since there is no background, and we thus focussed on analysing the simulated data.

As now detailed in a new SI section, we analysed the polariton field magnitude maps ($\text{abs}(E_z)$) to first extract the ray direction from circular cuts around the source, followed by exponential decay maps along those extracted directions, providing independently the intensity and the propagation lengths, defined as the distance until the initial amplitude of the field within a ray is reduced by a factor of $1/e$, for each ray as a function of twist angle. We show the results of this analysis below, as well as in the revised SI. Quite amazingly, we find a different behaviour for the two systems we considered in our work. For the homostructure, the ray suppression is found to be purely an amplitude effect, with no significant change in the propagation lengths. This suggests that the unidirectionality effectively emerges from a modulation of the effective mode density for each ray in the homostructure. In contrast, for the heterostructure we find a strict and very pronounced correlation between ray intensity and propagation length, further underlining the different physical origin of unidirectional ray formation in the two systems. We now include a short discussion of these mechanisms in the revised main text and included this new analysis in two new SI sections (S6 and S13).

Revised text in the manuscript:

Line 222:

Twisting the two layers reduces the intensity of one of the previously crossed rays towards its full suppression (Supplementary Fig. 6), enabling URPs."

Line 330:

"Notably, and in contrast to the homostructure, the twist-induced asymmetry for the heterostructure not only modulates the intensity but also its propagation lengths of the rays (Supplementary Fig. 15), with values ranging from $1 \mu\text{m}$ for the most suppressed rays to $5 \mu\text{m}$ for the most enhanced ones. This finding underlines a different physical origin of unidirectional ray formation in the two systems."

Line 450:

"Notably, the different mechanisms for URP formation between the homostructure and the heterostructure results in different twist-induced effects on the propagation lengths for each case, as detailed in Supplementary Figs. S6 and S15."

Analysis of the intensity and propagation lengths for the homostructure $\text{MoO}_3/\text{MoO}_3$ (left) and the heterostructure MoO_3/bGO (right). For the homostructure, the asymmetric response of the rays is largely an intensity effect with insignificant changes in the propagation length. For the heterostructure, on the other hand, intensity and propagation length are perfectly correlated for each ray. A full discussion is found in the revised SI (S6 and S13).

6. Line 284-282 provide an interpretation of the physical origins of unidirectional propagation, which I find to be a pivotal aspect of the study. To further strengthen this key point, I suggest that the authors include visualizations from 3D COMSOL simulations. Specifically, resending the vertical cross-section of the electric field profile along the radial direction at various angles would greatly enhance the reader's understanding of the phenomenon.

We thank the reviewer for this insightful comment. We calculated vertical cross sections as suggested for various twist angles along selected directions, see below. Despite the power of 3D simulations, these cuts do not necessarily reveal the nature of unidirectional ray emergence, unfortunately. Also, the bulk rays in the thick bottom layer that are visible in these cuts do not necessarily help in understanding the features of the unidirectional rays polaritons (they do not contribute to the modes of interest) and thus are expected to confuse rather than enlighten readers. Therefore, we would rather not include these calculations in the revised manuscript.

7. Line 327 addresses the propagation lengths of the rays, yet the term is not adequately defined within the manuscript. The concept of propagation length for ray polariton may be complex, as it would encompass decay due to material loss or the dispersion of electromagnetic energy across a finite energy. To provide clarity, I recommend that the author offers a comprehensive definition of the propagation length specific to Ray polaritons. Additionally, detailing the extraction process used to determine this value would greatly benefit the reader's understanding.

Again, we thank the reviewer for requesting a more quantitative analysis of our data. As detailed in the response to question nr. 5, we now include a full analysis of the ray intensities and propagation lengths from the simulations in the revised SI. This now also allows us to quantify the propagation lengths for both systems, homostructure and heterostructure. Indeed, the propagation length for the homostructure is $\sim 2 \mu\text{m}$ independent of the twist angle, whereas for the heterostructure, the propagation lengths vary between $\sim 1 \mu\text{m}$ for the most suppressed ray and $> 4 \mu\text{m}$ for the most enhanced ray. This analysis, with a proper definition of propagation length concept, is now included in the SI (S6 and S13 for the homostructure and heterostructure, respectively), and we revised the main manuscript accordingly as detailed in the response to question nr. 5.

Reviewer #3 (Remarks to the Author):

The authors reported on the unidirectional propagation of phonon polaritons in van der Waals stacks composed of homo- and heterostructures. They demonstrated that by forming homostructures of α -MoO₃ and heterostructures of α -MoO₃ and β -Ga₂O₃, the unidirectional flow of phonon polaritons can be realized by either tuning the excitation frequency or controlling the tuning angles. In stark contrast to previous studies on twisted bilayer α -MoO₃, where the thicknesses of the two α -MoO₃ layers are very thin and comparable to each other, the steering of phonon polaritons in the current study is achieved in stacks with very different thicknesses of their constituents. The underlying mechanisms governing such steering are discussed and show good agreement with the experimental findings.

Steering of phonon polaritons in the basal plane of van der Waals polar crystals is crucial for their applications in subwavelength waveguiding. The concept of twisted optics, which stacks two van der Waals slabs perpendicularly with the optical axes mismatched between the top and bottom constituents, has already proven to be an efficient approach. **In contrast to previous studies, the authors proposed a more facile method to construct the twisted structure, requiring only one thin van der Waals layer as the top layer.** In this way, the steering of polaritons occurs due to the modulation of the polariton propagation either by the dielectric environment provided by the thick bottom layer or by the hybridization of polaritons from both layers, depending on the excitation frequency employed. I find the results impressive and believe they are suitable for publication in Nature Communications. Here are a few concerns on the scientific points that may help improve the presentation of their study.

We thank the reviewer for their very positive assessment of our work. Please find the point-by-point response to the technical comments below.

1) The most significant innovation of this work is the use of vertically stacked van der Waals crystals with different thicknesses to achieve unidirectional propagation of phonon polaritons. The authors have indeed observed this phenomenon experimentally. However, they need to discuss the transformative breakthroughs this method could bring to practical applications, especially in comparison to previously reported results.

We thank the reviewer for this useful comment. There are two significant components of our work that distinguish it from previous approaches. First, as pointed out also by the reviewer, the possibility of steering the nanoscale light propagation by twisting a single vdW layer on a thick substrate may allow for a much easier device fabrication process compared to previous reports using double or triple layers. Secondly, the ray-like propagation with constant phase along the propagation direction is expected to be – quite similar to epsilon-near-zero materials – less sensitive to defects and obstacles, enabling more robust operation of on-chip photonic structures. We included some additional text in the revised manuscript to highlight these benefits.

Changes in the revised manuscript, line 97:

“Consequently, only one thin vdW layer is required to fabricate these stacks which constitutes a significant simplification with respect to previous reports of unidirectional PHP propagation where double or triple thin layers were used.”

2) In Figure 2 and 4, it is suggested to provide the dependence of unidirectional flow direction on the stacking angle (with fixed excitation frequency) and excitation frequency (with fixed stacking angle).

In response to reviewer 2, we have now included a quantitative analysis of ray direction, intensity, and propagation length in the SI figures S6 and S15 for homostructure and heterostructure, respectively, for a fixed frequency. Using the same analysis, we now also provide a plot of the ray directions as a function of frequency at a fixed stacking angle.

3) The thickness of the top layer is indeed much thinner than that of the bottom layer. I would like to know what would happen if the thickness of the top layer is reduced to a few atomic layers or even a single atomic layer.

We thank the reviewer for this very interesting question. Even though not directly related to this current work, the question of hyperbolicity in polaritonic materials like hBN, α -MoO₃, and β -Ga₂O₃ in the limit of extremely thin films is a highly debated one in general. Early work on hBN (Dai et al., Science 2014) has shown hyperbolic modes exist down to 4 monolayers. It is generally expected that the out-of-plane modes vanish in the monolayer limit (since LO and TO become degenerate) such that hyperbolicity would also disappear. However, rigid experimental proof or theoretical analysis is still pending to the best of our knowledge.

For our homostructure and heterostructure system, we performed transfer matrix simulations to estimate the effect of reducing the top layer thickness from 100 nm down to ~ 10 nm, see below. We find that for both systems, reducing the top layer thickness too much is detrimental to the unidirectional ray performance. For the homostructure, the increasing confinement of the thin film hyperbolic volume modes reduces the hybridization with low-confinement surface modes of the thick bottom layer, essentially decoupling both systems. For the heterostructure, the system is eventually dominated by the substrate surface polaritons if the top α -MoO₃ layer is getting optically too thin, thereby forfeiting any twist-control of the propagation.

Transfer matrix simulations of the isofrequency maps for the homostructure for $d_{\text{MoO}_3, \text{top}} = 100, 50, \text{ and } 10 \text{ nm}$ thickness at a frequency of $\omega = 900 \text{ cm}^{-1}$. For the thinnest top layer, hybridization between the low-confinement modes of the bottom layer and the high-confinement modes of the top layer is suppressed, leading to a symmetric map with vanishing unidirectionality.

Transfer matrix simulations of isofrequency surfaces of the heterostructure for $d_{\text{MoO}_3, \text{top}} = 100, 50,$ and 10 nm thickness at a frequency of $\omega = 712 \text{ cm}^{-1}$. For the thinnest top layer, the propagation within the $\alpha\text{-MoO}_3$ layer is pushed to very high momentum values, and in turn, the shear surface mode of the bGO becomes dominant, thus losing any twist-tuning capability of the unidirectional propagation.

4) In the discussion above Figure 2, the authors mention that the propagation direction of phonon polaritons can be actively controlled. However, the current experiments are actually based on specific frequencies and structures, and do not achieve reversible control. It is recommended to temper such description.

We thank the reviewer for pointing out this issue. Even though there have been several experimental demonstrations of active control of the twist angle in bilayer systems with sophisticated experimental arrangements, one might still question to what extent this qualifies as “active” control, in particular in the context of photonic device fabrication. We removed “active” in the revised manuscript to avoid any confusion.

5) In discussion above Figure 1, the authors state that “the other branch is suppressed completely”. This is not entirely accurate, as there is still energy flow along the weaker branch. The authors may consider modifying this description.

We have revised the respective text, now writing “the other branch is suppressed almost completely”.

REVIEWERS' COMMENTS

Reviewer #2 (Remarks to the Author):

I appreciate the author's positive response to my comments. The manuscript has significantly improved in the current version, and I am pleased with the revision. I would recommend the manuscript for publication in its present form.

Reviewer #3 (Remarks to the Author):

The authors have addressed the comments and questions raised by the reviewers well. I therefore recommend to accept their manuscript without further technical points.